

# Peripheral blood bovine lymphocytes and MAP show distinctly different proteome changes and immune pathways in host-pathogen interaction

Kristina J.H. Kleinwort[1], Stefanie M. Hauck[2], Roxane L. Degroote[1], Armin M. Scholz[3], Christina Hölzel[4,5], Erwin P. Maertlbauer[5] and Cornelia Deeg[1]

[1] Chair of Animal Physiology, Department of Veterinary Sciences, LMU Munich, Munich, Germany
[2] Research Unit for Protein Science, Helmholtz Zentrum Munich, German Research Center for Environmental Health GmbH, Munich, Germany
[3] Livestock Center of the Faculty of Veterinary Medicine, LMU Munich, Oberschleissheim, Germany
[4] Institute of Animal Breeding and Husbandry, Faculty of Agricultural and Nutritional Sciences, CAU Kiel, Kiel, Germany
[5] Chair of Hygiene and Technology of Milk, Department of Veterinary Sciences, LMU Munich, Oberschleissheim, Germany

Corresponding author
Cornelia Deeg,
Cornelia.Deeg@lmu.de

## ABSTRACT

*Mycobacterium avium* subsp. *paratuberculosis* (MAP) is a pathogen causing paratuberculosis in cattle and small ruminants. During the long asymptomatic subclinical stage, high numbers of MAP are excreted and can be transmitted to food for human consumption, where they survive many of the standard techniques of food decontamination. Whether MAP is a human pathogen is currently under debate. The aim of this study was a better understanding of the host-pathogen response by analyzing the interaction of peripheral blood lymphocytes (PBL) from cattle with MAP in their exoproteomes/secretomes to gain more information about the pathogenic mechanisms of MAP. Because in other mycobacterial infections, the immune phenotype correlates with susceptibility, we additionally tested the interaction of MAP with recently detected cattle with a different immune capacity referred as immune deviant (ID) cows. In PBL, different biological pathways were enhanced in response to MAP dependent on the immune phenotype of the host. PBL of control cows activated members of cell activation and chemotaxis of leukocytes pathway as well as IL-12 mediated signaling. In contrast, in ID cows CNOT1 was detected as highly abundant protein, pointing to a different immune response, which could be favorable for MAP. Additionally, MAP exoproteomes differed in either GroEL1 or DnaK abundance, depending on the interacting host immune response. These finding point to an interdependent, tightly regulated response of the bovine immune system to MAP and vise versa.

## INTRODUCTION

*Mycobacterium avium* subsp. *paratuberculosis* (MAP) is a critical pathogen for cattle and small ruminants, causing paratuberculosis with decreased milk production and in some animals, excessive loss of weight (*Yamamoto et al., 2018*). Paratuberculosis, also referred to as Johne's disease, is endemic world-wide endemic; no country or region has been found to be free of this disease (*Li et al., 2016*). Affected ruminants go through a long asymptomatic subclinical phase in which infection cannot reliably be detected by standard diagnostic tests (*Hobmaier et al., 2019*; *Li et al., 2017*). These subclinically infected animals can already shed or harbor MAP and thereby contaminate dairy products or meat (*Sweeney, 2011*). Diseased animals were shown to shed high numbers of MAP (*Machado et al., 2018*). Since viable MAP were found in pasteurized milk (*Gerrard et al., 2018*), dried dairy products like powdered infant formula (*Botsaris et al., 2016*) and in raw fermented sausages (*Lorencova et al., 2019*), MAP could be considered as possible foodborne pathogen.

A similar pathology in the intestinal tissue of patients with intestinal tuberculosis and paratuberculosis was described more than a century ago (*Dalziel, 1913*). Recently, an association between MAP and Crohn's disease was shown, initiating a discussion about a possible relationship of MAP in Crohn's pathogenesis (*Alcedo, Thanigachalam & Naser, 2016*). Johne's and Crohn's disease share clinical and histopathological similarities, MAP can survive standard pasteurization procedures and MAP antibodies can be detected in Crohn's patients, where macrolide antibiotics ameliorate disease (*Kuenstner et al., 2017*). In contrast, genotypes of MAP isolated from cattle and man are different, there is a lack of evidence for uptake of contaminated food in respective patients and MAP cannot consistently be isolated from Crohn's disease patients (*Mendoza, Lana & Díaz-Rubio, 2009*). Although MAP was detected widespread in many farms and different countries, the incidence of Johne's disease in ruminants is marginal (*Sergeant et al., 2019*). Bacteria can survive for 2–10 years without causing obvious symptoms of infection in cows (*Hermon-Taylor, 2000*). As seen in cattle farms, susceptibility to MAP infection differs in human populations (*Eslami et al., 2019*). This points to a complex disease in which several pathogens, environmental factors and an inappropriate immune response in genetically susceptible hosts participate in the cause of disease (*Eslami et al., 2019*). Since an enhanced susceptibility of the host contributes to pathogenesis in other mycobacteria associated diseases (e.g., tuberculosis) (*Scriba, Coussens & Fletcher, 2017*), our aim wasto gain further information about the interplay of MAP with the immune system in hosts with different immune capacities. This is also of interest because in cattle, the MAP eradication programs that are solely based on hygiene management are not very successful (*McAloon et al., 2019*). This could indicate certain reservoir cows that host and spread MAP without developing any clinical signs.

Little is known about the host-pathogen interaction of MAP and the immune system of its hosts (*Davis, 2015*). Functional differences in these responses could lead to aberrant reactions in susceptible hosts (*Davis, 2015*).

Recently, we detected a functionally different immune capacity in 22 % of cows from different herds in Germany using differential proteome analyses (*Lutterberg et al., 2018*). These cows differ in their constitutive immune proteome and they regulate different master immune regulators upon polyclonal immune cell stimulation. The phenotype is functionally correlated with an increased prevalence of mastitis, indicating an impact on the ability to fight infections (*Lutterberg et al., 2018*). Since the immune capacity of these cows clearly differs but the functional impact has to be characterized more accurately in future studies, we designated them immune deviant (ID) cows.

All living microorganisms are exposed to changing environmental parameters that define their habitats. Bacteria sense environmental changes and react to it with various stress response mechanisms (*Guo & Gross, 2014*). To gain information about the pathogenic mechanisms of MAP and how they respond to different immune response signals of their hosts, we co-incubated MAP with primary peripheral blood derived leukocytes (PBL) of control and ID cows. Since the aim of this study was a better understanding of the host-pathogen response, we analyzed the changes in the exoproteomes of MAP and the bovine peripheral blood derived lymphocytes.

The term exoproteome describes the protein content that can be found in the extracellular proximity of a given biological system (*Armengaud et al., 2012*). These proteins arise from cellular secretion, other protein export mechanisms or cell lysis, but only the most stable proteins in this environment will remain abundant (*Armengaud et al., 2012*). These proteins play roles in the organism's survival in extreme habitats such as saline environments (*Rubiano-Labrador et al., 2015*). Investigating the exoproteome of the pathogen and the secretome of PBL provides expanded coverage of the repertoire of proteins secreted in the stress response of MAP to different immune responses and this is essential for understanding respective mechanisms. Accordingly, this study aimed at providing a better understanding of the interplay of MAP with the immune system at the proteome level to identify the complex network of proteins involved in host-foodborne bacteria communication.

## MATERIALS AND METHODS

### *Mycobacterium avium* subsp. *paratuberculosis*

The bacterial strain used in this study was *Mycobacterium avium* subsp. *paratuberculosis* (MAP) (DSM 44133), purchased from German Collection of Microorganisms and Cell Cultures (DSMZ, Braunschweig, Germany). MAP were grown on Herrold's egg yolk agar (HEYM) (BD Biosciences, Heidelberg, Germany) for 4 weeks prior to harvesting for the co-incubation experiment. MAP were yielded through rinsing the agar of the cultivation tubes with phosphate buffered saline (PBS) and gentle scratching.

### PBL isolation of control and immune deviant cows

Blood samples of cows were collected in tubes supplemented with 25.000 I.U. heparin. Blood was then diluted 1:2 with PBS pH 7.2 and subsequently layered on density gradient separating solution (Pancoll; PanBiotech, Aidenbach, Germany). After density gradient centrifugation (room temperature, 290×*g*, 25 min, brake off), PBL were obtained from at
the intermediate phase. Cells were then washed 3× in PBS (4 °C). Withdrawal of blood was permitted by the local authority Regierung von Oberbayern, Munich, Permit No. 55.2-1-54-2532.3-22-12. For determination of control or ID status, PBL were tested in in vitro proliferation assays as described in *Lutterberg et al. (2018)*. Respective animals were tested at least 11 times, before being assigned to control or ID status. The animals were from a MAP-free farm and were tested negative for MAP antibodies and no MAP was detected in culture from feces.

## Co-incubation of MAP with primary bovine PBL

Primary bovine PBL ($2 \times 10^7$) of control and ID cows were cultivated in six well plates in three ml RPMI each. The same number of live MAP ($2 \times 10^7$; MOI = 1) were added for 48 h to one well, control wells were not infected. After 48 h, three technical replicates per experiment were centrifuged for 10 min at $350 \times g$ and the supernatants were filtered through 0.5 μm filters (=exoproteome) and stored at $-20$ °C until filter aided sample preparation (FASP). Cell pellets were lysed and total protein content measured colorimetric with the Bradford assay (*Bradford, 1976*).

## Proteolysis and LC-MS/MS mass spectrometry

A total of 10 μg total protein was digested with LysC and trypsin by FASP as described in *Grosche et al. (2016)*. Acidified eluted peptides were analyzed in the data-dependent mode on a Q Exactive HF mass spectrometer (Thermo Fisher Scientific, Bremen, Germany) online coupled to a UItimate 3000 RSLC nano-HPLC (Dionex). Samples were automatically injected and loaded onto the C18 trap column and after 5 min eluted and separated on the C18 analytical column (75 μm IDx15 cm, Acclaim PepMAP 100 C18. 100 Å/size; LC Packings, Thermo Fisher Scientific, Bremen, Germany) by a 90 min non-linear acetonitrile gradient at a flow rate of 250 nl/min. MS spectra were recorded at a resolution of 60,000 and after each MS1 cycle, the 10 most abundant peptide ions were selected for fragmentation.

## Protein identification and label-free quantification

Acquired MS spectra were imported into Progenesis software (Version 2.5 Nonlinear Dynamics, Waters) and analyzed as previously described (*Hauck et al., 2012*; *Hauck et al., 2017*). After alignment, peak picking, exclusion of features with charge state of 1 and >7 and normalization, spectra were exported as Mascot Generic files and searched against a database containing all entries of *Mycobacterium avium* subspecies *paratuberculosis* from NCBI Protein database combined with the Ensembl bovine database (Version 80) with Mascot (Matrix Science, Version 2.5.1). Search parameters used were 10 ppm peptide mass tolerance, 20 mmu fragment mass tolerance, one missed cleavage allowed, carbamidomethylation set as fixed modification and methionine oxidation and deamidation of asparagine and glutamine as variable modifications. Mascot integrated decoy database search was set to a false discovery rate (FDR) of 1% when searching was performed on the concatenated mgf files with a percolator ion score cutoff of 13 and an appropriate significance threshold p. Peptide assignment was reimported to Progenesis

Software. All unique peptides allocated to a protein were considered for quantification. Proteins with a ratio of at least five-fold in normalized abundance between control and ID samples were defined as differentially expressed.

### Enriched pathway analyses

Abundances of the identified proteins were defined as differentially expressed based on the threshold of protein abundance ratio and their assignment to MAP or bovine exoproteome. Venn diagram was made with open source tool: http://bioinformatics. psb.ugent.be/webtools/Venn/. The protein–protein interaction network of differentially-accumulated proteins was analyzed with GeneRanker of Genomatix Pathway System software (Version 3.10; Intrexon Bioinformatics GmbH, Munich, Germany; settings: Orthologous genes from *H. sapiens* were used for this ranking, species bovine, analyzed were proteins with ≥5-fold change). Hierarchical clustering and enrichment analysis of biological pathways were conducted with open source software ShinyGO v0.50: http://ge-lab.org/go/ (*Ge & Jung, 2018*), *p*-value cutoff FDR was set to 0.05. ShinyGO uses gene ontology (GO) annotation and gene ID mapping of animal and plant genomes in Ensembl BioMart and in addition archaeal, bacterial and eukaryotic genomes based on STRING-db v10. Additional pathway data are collected for some model species from difference sources. Since many GO terms are related or even redundant, the relatedness was visualized with the hierarchical clustering tree and network, where related GO terms are grouped together based on how many genes they share. The size of the solid circle corresponds to the enrichment FDR.

In the network analysis, each node represented an enriched GO term. Related GO terms were connected by a line, whose thickness reflects percent of overlapping genes. Size of the node corresponded to number of genes.

## RESULTS

### Differentially abundant proteins in secrotomes of control and immune deviant bovine peripheral blood lymphocytes in response to co-incubation with MAP

We investigated the immune response of primary PBL isolated from control and immune deviant (ID) animals. After 48 h of co-incubation with live MAP, we harvested the mixed secretomes and exoproteomes of PBL and MAP and analyzed them with mass spectrometry. Overall, we identified 826 proteins (811 bovine and 15 MAP proteins). Cluster analysis confirmed significant differences in protein abundances between PBL of control and ID cow in host-pathogen response (Fig. 1). In secretome of control cow, 90 proteins were differentially upregulated (Fig. 1, ≥5-fold change of expression, blue circle) in contrast to 38 proteins that were higher abundant in ID cow (Fig. 1, light red circle).

### Different biological pathways were regulated in immune deviant versus control bovine PBL in response to MAP

Interpretation of large protein sets can be performed through enrichment analyses, using published information for examination of overrepresentation of a known set of genes

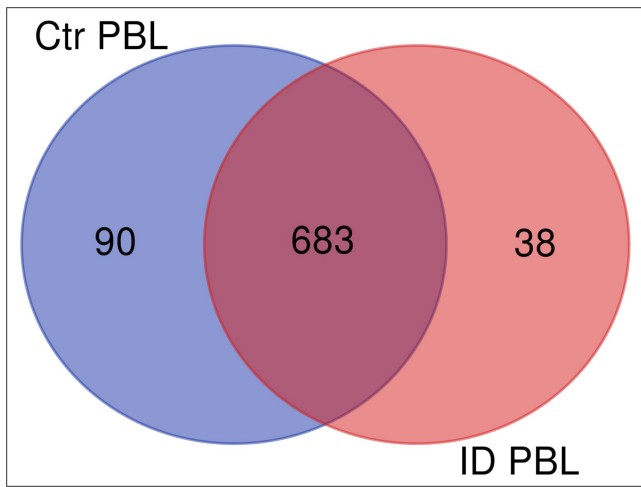

**Figure 1 Overlap (Venn diagram) of differentially (≥5 fold) expressed proteins between secretomes/ exoproteomes of control cow (blue) and ID cow (red).** From a total of 811 identified proteins, 90 were higher abundant in control and 38 in ID.

within the input gene list (*Ge & Jung, 2018*; *Merico et al., 2010*). Since many GO terms are related or redundant, we used a hierarchical clustering tree and network analysis (ShinyGO v.050 (*Ge & Jung, 2018*)). The top regulated biological process pathways in the secretome/excretome of the control were all related to RNA splicing (Fig. 2A). The top regulated immune pathways were cellular immune responses of various leukocyte subsets, chemotaxis and the reaction to IL-12 (Fig. 2A).

In contrast, in the secretome/excretome of ID PBL, the top regulated pathways were response to stress and immune system process (Fig. 2B). The enriched immune pathways comprised distinct routes of immune response in ID PBL, focusing on closely related positive regulations of cellular functions, complement activation and humoral immune response (Fig. 2B).

Visualization of network clearly indicates two major regulated networks in control PBL's secretome (Fig. 3A) and one distinct, major enriched network in ID proteins (Fig. 3B).

## GroEI1 and DnAK were differentially abundant in MAPs interacting with different hosts

In the MAP exoproteome, although small in overall numbers, different protein abundances were also detectable. The two most notable were the protein GroE11, upregulated in MAP co-incubated with control PBL, versus the protein DnaK, which was upregulated in MAP co-incubated with ID PBL. Other MAP proteins were differentially up or down regulated, but not to as great an extent as GroE11 or DnaK, as noted in Table 1.

## DISCUSSION

*Mycobacterium avium* subsp. *paratuberculosis* could be a foodborne pathogen and has been discussed in association with several human diseases (*Cossu et al., 2019*; *Pierce, 2018*; *Sechi & Dow, 2015*). MAP has been identified I in patients with inflammatory bowel

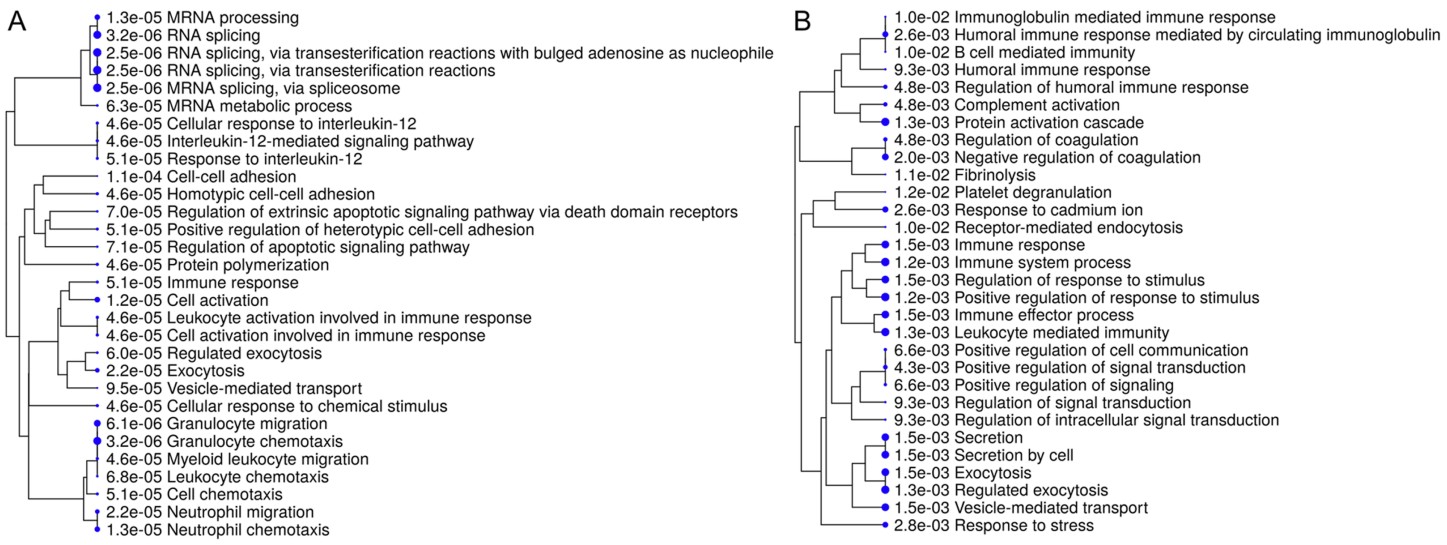

**Figure 2 Hierarchical clustering tree and network of related GO terms of differentially expressed proteins in secretomes/exoproteomes of control PBL (A) and ID PBL (B) illustrates marked differences.** GO terms are grouped together based on how many genes they share. The size of the solid circle corresponds to the enrichment false discovery rate.

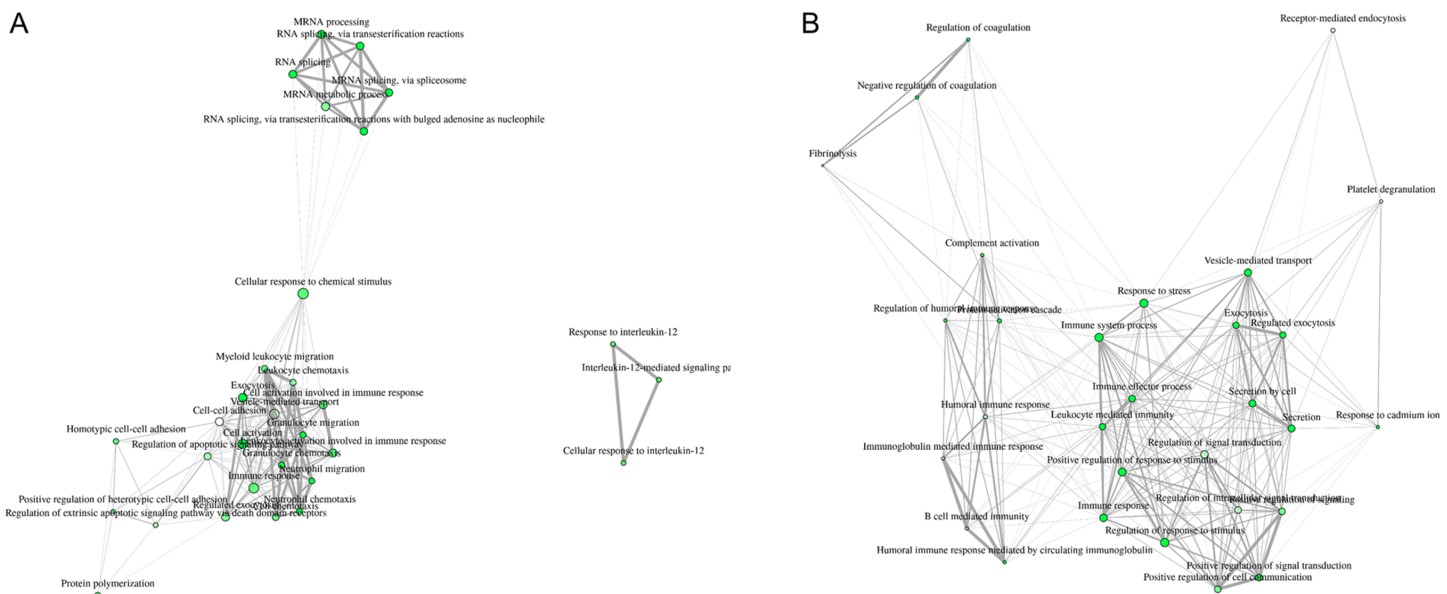

**Figure 3 Visualization of overlapping relationships among enriched gene-sets revealed (A) two major networks as shown by network view for enriched GO molecular component terms in control PBL after MAP infection and (B) in ID PBL after MAP infection.** Related GO terms are connected by a line whose thickness reflects percent of overlapping genes. Size of the nodes corresponds to number of genes.

diseases like Crohn's disease and ulcerative colitis, as well as autoimmune diseases like type one diabetes, multiple sclerosis, rheumatoid arthritis and Hashimoto's thyroiditis (*Garvey, 2018*), but so far, a causal association has not yet been proven in any of these cases. Since MAP can survive many of the standard techniques of food decontamination (e.g., pasteurization), they are regularly found alive in pasteurized milk (*Gerrard et al., 2018*) and in dried dairy products such as powdered infant formula (*Botsaris et al., 2016*). If MAP

**Table 1  Regulation of MAP proteins identified in secretomes/exoproteomes after co-incubation with control and ID PBL.**

| | Accession | Gene symbol | Description | Ratio MAP Ctr/ID |
|---|---|---|---|---|
| 1 | AAS06815 | GroEL1 | pep:novel chromosome:GCA_000007865.1:Chromosome:4733338:4734954:1 gene: MAP_4265 transcript:AAS06815 description:"GroEL1" | 6.07 |
| 2 | ETB04387 | F0F1 ATP synthase subunit alpha | pep:novel supercontig:GCA_000504785.1:contig000131:9486:11150:1 gene: O979_07265 transcript:ETB04387 description:"F0F1 ATP synthase subunit alpha" | 3.26 |
| 3 | ETB05569 | DNA-binding protein | pep:novel supercontig:GCA_000504785.1:contig000072:8572:9225:1 gene: O979_04085 transcript:ETB05569 description:"DNA-binding protein" | 2.47 |
| 4 | AAS06727 | RplN | pep:novel chromosome:GCA_000007865.1:Chromosome:4652017:4652385:1 gene: MAP_4177 transcript:AAS06727 description:"RplN" | 2.34 |
| 5 | AAS06486; ELP44387; ETB08817 | GroEL2 | pep:novel chromosome:GCA_000007865.1:Chromosome:4395922:4397547:1 gene: MAP_3936 transcript:AAS06486 description:"GroEL2" | 1.12 |
| 6 | AAS06693 | Tuf | pep:novel chromosome:GCA_000007865.1:Chromosome:4620946:4622136:1 gene: MAP_4143 transcript:AAS06693 description:"Tuf" | 0.88 |
| 7 | ETB02420 | enoyl-CoA hydratase | pep:novel supercontig:GCA_000504785.1:contig000238:14619:15410:1 gene: O979_11400 transcript:ETB02420 description:"enoyl-CoA hydratase" | 0.86 |
| 8 | ETB00930 | 2-isopropylmalate synthase | pep:novel supercontig:GCA_000504845.1:contig000411:16560:18359:-1 gene: O978_19245 transcript:ETB00930 description:"2-isopropylmalate synthase" | 0.75 |
| 9 | AAS06690 | RpsL | pep:novel chromosome:GCA_000007865.1:Chromosome:4617754:4618128:1 gene: MAP_4140 transcript:AAS06690 description:"RpsL" | 0.67 |
| 10 | AAS06681; ETB45918; ETB45924 | RpoC | pep:novel chromosome:GCA_000007865.1:Chromosome:4606729:4610679:1 gene: MAP_4131 transcript:AAS06681 description:"RpoC" | 0.62 |
| 11 | AAS06680 | RpoB | pep:novel chromosome:GCA_000007865.1:Chromosome:4603087:4606683:1 gene: MAP_4130 transcript:AAS06680 description:"RpoB" | 0.48 |
| 12 | AAS02778 | ClpC | pep:novel chromosome:GCA_000007865.1:Chromosome:488298:490829:1 gene: MAP_0461 transcript:AAS02778 description:"ClpC" | 0.46 |
| 13 | ETB04840 | GTP-binding protein YchF | pep:novel supercontig:GCA_000504785.1:contig000104:1736:2809:1 gene: O979_06015 transcript:ETB04840 description:"GTP-binding protein YchF" | 0.43 |
| 14 | ETB04389; AAS04768 | ATP synthase subunit beta | pep:novel supercontig:GCA_000504785.1:contig000131:12100:13557:1 gene: O979_07275 transcript:ETB04389 description:"ATP synthase subunit beta" | 0.26 |
| 15 | AAS06390 | DnaK | pep:novel chromosome:GCA_000007865.1:Chromosome:4295544:4297415:1 gene: MAP_3840 transcript:AAS06390 description:"DnaK" | 0.04 |

**Note:**
Regulation of MAP proteins identified in secretomes/exoproteomes after co-incubation with control and ID PBL.

organisms play a role in the pathogenesis of human diseases, we think that this must be associated with a certain type of susceptibility in these human hosts. MAP is ubiquitously found in milk and meat while only a minor proportion of consumers (if at all) is affected by MAP-associated diseases. The same is true for cows: while they are often in contact with MAP, resulting in high frequencies of seropositive animals—approximately 20% and at least 3–5% in several countries (*Boelaert et al., 2000*; *Nielsen & Toft, 2009*)—Johne's disease incidence is very rare. For example, a total of 232 clinical cases of Johne's disease were reported in Ireland from 1995 to 2002 (*Kennedy et al., 2014*), yielding an average annual rate of approximately 0.0005%, given a cattle population of six million (*Maher, Good & More, 2008*). A study examining environmental samples from 362 dairy

farms located in all 10 provinces of Canada for detection of MAP by culture revealed true prevalence estimates of 66% for farms in Western Canada, 54% in Ontario, 24% in Québec, and 47% in Atlantic Canada (*Corbett et al., 2018*).

It is known from infection with *Mycobacterium tuberculosis* (MTB), that there is a 10% probability that the host will develop active tuberculosis and the bacterium may invade multiple organs (*Zhai et al., 2019*). Although nine million new cases of active tuberculosis are still reported annually, an estimated one-third of the world is infected with MTB while remaining asymptomatic, defined as latent TB (*Dye et al., 1999*). Among the individuals with latent TB, only 5–10% will develop active tuberculosis disease in course of their lifetime, because they effectively control the infection through their immune response (*Dye et al., 1999*). This immune response after MTB infection is highly complex as the bacteria have intricate immune escape mechanisms (*Zhai et al., 2019*).

For MAP, little is known about host-pathogen reactions in general and whether different immune responses exist, but a thorough examination of the respective mechanisms is of major importance to obtain functional data that will allow a better understanding and a substantiated risk assessment. In the post-genomic era, proteomics represents a key discipline to perform in depth studies and identify the complex network of proteins involved in such host-bacteria communication. In this study, we analyzed the exoproteomes/secretomes of MAP and host cow PBL with known, functionally different immune capacities (*Lutterberg et al., 2018*) using differential proteome analyses after co-incubation in vitro. Interestingly, there were significant differences in protein abundances secreted from control and ID PBL. Ninety proteins were ≥5 fold higher abundant in control PBL after interaction with MAP. In the control, the top regulated immune pathways described cell activation and chemotaxis of leukocytes as well as IL-12 mediated signaling pathways. For MAP it was shown, that IL-12 transcription is increased in infected bovine macrophages within 6 h (*Weiss et al., 2002*), probably to enhance the developing T cell response (*Bannantine et al., 2015*). Later, the in vitro challenged monocytes-derived macrophages from healthy cows enhanced production of the anti-inflammatory cytokine interleukin-10 (IL-10) as measured by qRT-PCR (*Abendaño, Juste & Alonso-Hearn, 2013*). This counteracted initial IL-12 production (*Abendaño, Juste & Alonso-Hearn, 2013*) and high levels of IL-10 in paratuberculosis were shown promote the survival of MAP (*Hussain et al., 2016*). Further, in THP1 cells, a human monocytic cell line, IL-12B gene expression was downregulated immediately after in vitro infection with MAP (*Motiwala et al., 2006*). To our knowledge, the role of IL-12 in MAP infections of cows has not been analyzed so far on protein level (*Abendaño, Juste & Alonso-Hearn, 2013*), but the IL-12 associated immune response should also be protective. In tuberculosis it was shown that the induction of protective IFN-γ T cell responses against primary *M. tuberculosis* infection clearly depends on IL-12 (*Khader et al., 2006*). Mice lacking IL-12p40 cannot control the growth of MTB bacterial infection (*Cooper et al., 1997*). These findings in mouse models were confirmed in man, where IL-12 was shown to be critical for preventing tuberculosis (*Alcaïs et al., 2005*). Therefore, we hypothesize that secretion of IL-12 indicates a protective immune response against MAP infection in control PBL. But, further investigation about the role of IL-12 in MAP infection is needed,
probably in naturally infected animals that cope differentially with clearing of MAP from their body.

In ID PBL on the other hand, CCR4-NOT transcription complex, subunit 1 (CNOT1) was detected as highly abundant protein (Table S1). This novel finding of CNOT1 regulation in the bovine immune system is very interesting, because CCR4-NOT complex members have recently been shown to function as regulators ensuring repression of the MHC class II locus in human cell lines (*Rodríguez-Gil et al., 2017*). Poor MHC class II expression can cause autoimmune or infectious diseases, since MHCII is indispensable for adequate immune responses (*Rodríguez-Gil et al., 2017*). Additionally, CNOT proteins also contribute to the downregulation of MHC class I gene expression by influencing transcription and mRNA degradation (*Yang et al., 2016*). There is evidence that mycobacteria interfere with the synthesis and expression of MHC class I and II molecules and their participation in antigen processing as a mechanism of immune evasion (*Harding & Boom, 2010*; *Noss, Harding & Boom, 2000*; *Tufariello, Chan & Flynn, 2003*). Experimentally MAP exposed Merino sheep that were monitored during course of disease revealed an important role for MHC class I and II gene expression and disease outcome (*Purdie et al., 2019*). There was a prolonged overall inhibition of expression of MHC class I genes the multibacillary and the paucibacillary cohorts, promoting enhanced survival of the intracellular pathogen MAP in infected sheep (*Purdie et al., 2019*). A modulation of MHC genes was also shown in a transcriptomic analysis of experimentally infected Holstein–Friesian calves (*David et al., 2014a, 2014b*) and very early differential regulation of MHC genes in young, MAP exposed cattle (*Purdie et al., 2012*).

Further investigation is needed to analyze a possible connection of CNOT1 to the immune evasion mechanisms happening in MAP infections, especially since it is not known so far, whether the alterations to MHC expression is driven by the pathogen or by the host (*Purdie et al., 2012*).

Enriched network analyses of secreted proteins from ID PBL revealed further major different functions of the 38 differentially abundant proteins (Fig. 3B). Members of complement activation pathway were enriched in ID PBL. We think the regulated candidates in ID PBL merit further investigation in future studies to clarify whether they indicate an immune response in favor of MAP infection or just another way to successfully fight mycobacterial infections.

Interestingly, in the MAP exoproteome (although small in overall numbers) different protein abundances were also detectable. Exoproteomes arise from cellular secretion, other protein export mechanisms or cell lysis, but only the most stable proteins in this environment will remain abundant (*Armengaud et al., 2012*). In MAP co-incubated with control PBL, GroEL1 showed 6-fold higher abundance. GroEL1 belongs to the family of 60 kDa heat shock proteins, also known as Cpn60s (GroELs) which are components of the essential protein folding machinery of the cell, but are also dominant antigens in many infectious diseases (*Sharma et al., 2016*). GroEL1 from MAP is highly immunogenic for cows (*Yang et al., 2016*). The exact function of GroEL1 in MAP is not clarified so far, but it is highly similar to the respective protein in MTB (Rv3417c), where GroEL1 is important for bacterial survival under low aeration by affecting the expression of genes

known for hypoxia response (*Sharma et al., 2016*). From co-cultivation of MAP with ID PBL, on the other hand, DnaK emerged as differentially abundant protein. DnaK is a HSP70 family chaperone protein with essential function in stress induced protein refolding and DnaK loss is accompanied by disruption of membrane structure and increased cell permeability (*Fay & Glickman, 2014*). DnaK is essentially required for cell growth in mycobacteria due to a lack of redundancy with other chaperone systems (*Fay & Glickman, 2014*). This finding from MAP to ID PBL co-cultivation points to regulation of important survival mechanisms in MAP. However, these results must be interpreted with care because our analyzed MAP exoproteome only comprised 15 proteins.

Our data provide novel information about MAP-leukocyte interaction, adding to a more comprehensive picture of host-pathogen interactions. Co-incubation of MAP with cells from animals with different immune capacities led to significant differences in PBL secretomes and different immunological pathways enhanced in the hosts. In exoproteomes of respective MAPs, GroEL1 and DnaK were differentially abundant. These analyses gave a deeper insight into the different responses of host PBL and MAP bacteria.

## CONCLUSIONS

In this study, several novel proteins were identified with changed abundance in host-pathogen interaction. These candidates merit further investigations in the future to clarify their functional role in infection control.

## ABBREVIATIONS

| | |
|---|---|
| **CCR4** | CCR4-NOT transcription complex, subunit 1 |
| **FASP** | Filter-aided sample preparation |
| **ID** | Immune deviant |
| **IL** | Interleukin |
| **HEYM** | Herrold's egg yolk agar |
| **MAP** | *Mycobacterium avium* subsp. *paratuberculosis* |
| **MTB** | *Mycobacterium tuberculosis* |
| **PBL** | Peripheral blood derived lymphocytes |
| **PBS** | Phosphate buffered saline |

## ACKNOWLEDGEMENTS

The authors would like to thank Barbara Amann for excellent technical support.

### Funding

The IGF Project 18388 N of the FEI was supported via AiF within the program for promoting the Industrial Collective Research (IGF) of the German Ministry of Economic Affairs and Energy (BMWi), based on a resolution of the German Parliament. The funders

had no role in study design, data collection and analysis, decision to publish, or preparation of the manuscript.

## Grant Disclosures
The following grant information was disclosed by the authors:
The IGF Project 18388 N of the FEI was supported via AiF within the program for promoting the Industrial Collective Research (IGF) of the German Ministry of Economic Affairs and Energy (BMWi), based on a resolution of the German Parliament.

## Competing Interests
The authors declare that they have no competing interests.

## Author Contributions
- Kristina J.H. Kleinwort performed the experiments, analyzed the data, prepared figures and/or tables, authored or reviewed drafts of the paper, approved the final draft.
- Stefanie M. Hauck performed the experiments, analyzed the data, contributed reagents/ materials/analysis tools, authored or reviewed drafts of the paper, approved the final draft.
- Roxane L. Degroote analyzed the data, prepared figures and/or tables, authored or reviewed drafts of the paper, approved the final draft.
- Armin M. Scholz performed the experiments, contributed reagents/materials/analysis tools, authored or reviewed drafts of the paper, approved the final draft.
- Christina Hölzel performed the experiments, contributed reagents/materials/analysis tools, authored or reviewed drafts of the paper, approved the final draft.
- Erwin P. Maertlbauer analyzed the data, contributed reagents/materials/analysis tools, authored or reviewed drafts of the paper, approved the final draft.
- Cornelia Deeg conceived and designed the experiments, analyzed the data, contributed reagents/materials/analysis tools, prepared figures and/or tables, authored or reviewed drafts of the paper, approved the final draft.

## Animal Ethics
The following information was supplied relating to ethical approvals (i.e., approving body and any reference numbers):

Regierung von Oberbayern, Munich, approved this research (permit no. 55.2-1-54-2532.3-22-12).

## Data Availability
The raw proteomics data are available in Table S1.

## Supplemental Information
Supplemental information for this article can be found online at http://dx.doi.org/10.7717/peerj.8130#supplemental-information.

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
