# Peer review of "Peripheral blood bovine lymphocytes and MAP show distinctly different proteome changes and immune pathways in host-pathogen interaction"

_PeerJ, doi:10.7717/peerj.8130_

## Round 0.1 · original submission · Major Revisions

A number of doubts or concerns were raised by reviewers, particularly, I consider necessary to clarify those raised by reviewer 2, so that your work gets an increased chance of getting understood by potential readers.

·

Basic reporting

I have made several minor suggestions in my annotated pdf. Overall the writing is very clear and succinct, and was a pleasure to read.

Experimental design

the discussion of the design and results was unusually clear and well written.

Validity of the findings

Very interesting results. With the exception of the IL-12 issue, i have only minor suggested changes

Additional comments

an exceptionally well written and easy to understand article, thank you. All suggested changes are in my annotated comments. I may be mis-understanding the IL-12 issue in the prior literature, but you will have to clarify please!

·

Basic reporting

The manuscript described the proteomic analysis of the exoproteomes/secretomes of peripheral blood lymphocytes co-incubated with Mycobacterium avium subsp. paratuberculosis in both an immunocompetent and immunomodulated animal model. Although the manuscript has its merits, there are a number of issues that need to be addressed.
Please, find below a list of points related to each section of the manuscript.

Abstract
• L28-33: Your introductory part of the abstract needs more detailing, those lines were not enough justification for the studied point, there is a focus on MAP shedding while the conclusion line stating the lack of information regarding host-pathogen interplay. I suggest that you restate this part to make it more relevant to the study point. Also, in L30 the author stated that “can be transmitted to food”. Which food? is it food for human consumption or animal food. I suggest that the author should be more precise and state facts in more clear way.
• L37: up to this part of the manuscript, the reader is not familiar with the term immune deviant animal, and when checking the reference of that term (reference 20) it describes a cattle immunophenotype related to mastitis. I suggest changing this statement to “a different immune susceptibility cattle referred as immune deviant cattle”.
• L42: MAP reacted differently to the host, different in which way? Or differently in compared to what? I don’t think this is an accurate phrase here. I suggest removing this sentence and just stating directly “Additionally, MAP exoproteome differed ....”

Introduction
• L61: change it to “MAP could be considered as foodborne pathogens”
• L64: missing reference.
• L71-74: this statement is confusing and needs rewriting. The outcome is not related to the first sentence.
• L79: “We wanted” change it to “our aim is to gain ….”.
• L97-99: this statement needs restating. Stating the aim of the study here and restating it later is redundancy.

Material and Methods
• L113: “these experiments” change to “this study”
• L130: co-incubation is a better description.
• L136: the author just mentioned Bradford assay without reference or explanation. I suggest adding both.
• L161: I don’t understand “control and ID or samples”. Please restate to make it less confusing.
• L160-161 and L163-164: please clarify how in the first sentence the five-fold in normalized abundance make a protein is differentially expressed, while in the second one protein abundance ratio of ≥1.5. do both statements refer to different parameters? If so, please consider consistency of the referred parameter throughout the section.
• L170-171: please state clearly all the bioinformatic analyses done by ShinyGO.

Experimental design

Although the manuscript has an acceptable experimental design to answer the defined research question, there are a number of issues that need to be addressed.
Please, find below a list of points related to each section of the manuscript.

Technical issue
• L115-118: I don’t see a reason to grow MAP on solid media and collect it by rinsing and scratching the surface of the media with PBS, instead it could have been grown in liquid media and obtain a clear CFU count in order to calculate a clear MOI.
• L132: Strange quantification of the MAP bacteria used in this experiment. 15 ug is not the right measurement of an organism to be used in a co-incubation experiment, this weight could include unviable bacteria or random proteins from the media since there was scratching during the preparation of the bacterial suspension. Also, no multiplicity of infection (MOI) was mentioned which would question whether this experiment was done under optimal condition or not.
• L127: Can the author state the reason animals were tested at least 11 times for inclusion in the study? Why specifically 11 times.

Validity of the findings

Results
• L177: the term “in vitro infection” was used here, while previously the author used co-incubation without stating the reason for this change. Does it refer to a different aspect of the current infection? If not, please consider the consistency of the used term throughout the manuscript.
• L181: the in brackets; refer to the figure only using its number, no need for describing its nature.
• L184: “less” instead of “small”
• L191: the author didn’t clarify the description of the “we used a hierarchical clustering tree and network” is it a software or an analysis. Please finish the sentence.

Discussion
• L228: “From tuberculosis” is not a complete statement. Instead, use “As observed in tuberculosis infection”.
• L265-268: based on the previously mentioned findings, the author may have concluded and stated a better conclusion. I suggest rewriting this part with better statements that describe what is the possible effect of downregulating MHC class I and II and the upregulation of the complement fixation pathway on infection of an intracellular organism like MAP.

---

## Round 0.2 · accepted · Accept

Congratulations on the acceptance of your revised manuscript and thanks for taking into account the reviewers comments.

·

Basic reporting

no comment

Experimental design

no comment

Validity of the findings

no comment

Additional comments

The author made appropriate corrections based on my comments